# Comprehensive Metabolite Profiling of Chemlali Olive Tree Root Extracts Using LC-ESI-QTOF-MS/MS, Their Cytotoxicity, and Antiviral Assessment

**DOI:** 10.3390/molecules28124829

**Published:** 2023-06-17

**Authors:** Karim Toumi, Łukasz Świątek, Anastazja Boguszewska, Krystyna Skalicka-Woźniak, Mohamed Bouaziz

**Affiliations:** 1Laboratoire d’Electrochimie et Environnement, Ecole Nationale d’Ingénieurs de Sfax, Université de Sfax, BP 1173, Sfax 3038, Tunisia; toumikarim1996@gmail.com; 2Department of Virology with Viral Diagnostics Laboratory, Medical University of Lublin, 1 Chodzki Street, 20-093 Lublin, Poland; lukasz.swiatek@umlub.pl (Ł.Ś.); anastazja.boguszewska@umlub.pl (A.B.); 3Department of Chemistry of Natural Products, Medical University of Lublin, 1 Chodzki Street, 20-093 Lublin, Poland; kskalicka@pharmacognosy.home.pl; 4Institut Supérieur de Biotechnologie de Sfax, Université de Sfax, BP 1175, Sfax 3038, Tunisia

**Keywords:** olive roots, ultrasonic extraction, LC-MS, phenolic compounds, biological activities

## Abstract

The large quantity of olive roots resulting from a large number of old and unfruitful trees encouraged us to look for ways of adding value to these roots. For this reason, the current research work is devoted to the valorization of olive roots by identifying active phytochemicals and assessing their biological activities, including the cytotoxicity and antiviral potential of different extracts from the *Olea europaea* Chemlali cultivar. The extract, obtained by ultrasonic extraction, was analyzed using the liquid chromatography-mass spectrometry technique (LC-MS). The cytotoxicity was evaluated through the use of the microculture tetrazolium assay (MTT) against VERO cells. Subsequently, the antiviral activity was determined for HHV-1 (Human Herpesvirus type 1) and CVB3 (Coxsackievirus B3) replication in the infected VERO cells. LC-MS analysis allowed the identification of 40 compounds, classified as secoiridoids (53%), organic acids (13%), iridoids (10%), lignans (8%), caffeoylphenylethanoid (5%), phenylethanoids (5%),sugars and derivatives (2%), phenolic acids (2%), and flavonoids (2%). It was found that extracts were not toxic to the VERO cells. Moreover, the extracts did not influence the appearance of HHV-1 or CVB3 cytopathic effects in the infected VERO cells and failed to decrease the viral infectious titer.

## 1. Introduction

Tunisia is regarded as the most important olive-producing country in the southern Mediterranean region. More than 30% of the agricultural land proved to be devoted to cultivating olive trees. These trees are distributed from north to south, with very different bioclimatic conditions and a wide range of varieties. Today, Tunisia ranks the fourth country in the world for the number of cultivated olive trees, estimated at more than 86 million trees covering an area of 1,800,000 hectares [1]. In addition to oil as the basic product, the olive oil industry produces a large number of by-products, including pruning products, pomace, and margines. The valorization of these residues has become crucial in order to avoid increased contamination. Prunings display multiple uses, including animal feed [2] and compost production [3].

These residues are known for their beneficial properties for human health, referring to their richness in phenolic compounds, especially oleuropein [4]. These compounds exhibit, among others, antioxidant, anticancer, and antimicrobial potentials, which makes them very significant for the health and food industries [5,6]. The growing interest in phenolic compounds in olive oil permeates all olive products that can be consumed as food (table olives, olive paste) or that are generated as by-products of the olive industry [6].

Phenolic compounds are known for their intrinsic physiological roles in both the plant and animal kingdoms. Referring to their structure, phenolic compounds are able to bind to specific enzymes and proteins and thus modify enzymatic balances. In the plant kingdom, these substances are often involved in the defense mechanisms developed by plants against predation by insects and herbivores, infections, and numerous microbial aggressions. In addition, phenolic compounds are associated with different physiological processes, namely cell growth, differentiation, bud dormancy, and flowering [7,8]. As far as herbal medicine is concerned, the effects of various medicinal plants are attributed, in whole or in part, to the phenolic compounds existing in these plants. These substances have biological activities, making them beneficial to human health. Multiple studies emphasize that polyphenols can reduce the risk of a number of pathologies, particularly those related to aging and oxidative damage, such as cancer and cardiovascular or neurodegenerative diseases [9]. Thus far, the structural characterization of phenolic compounds has significantly improved thanks to the use of liquid chromatography coupled with mass spectrometry (LC-MS) [10,11]. Among the most prominent researchers, we mention Ammar et al. [12], who conducted a phytochemical study of olive wood by means of analyses based on mass spectrometry and its comparison with the by-products derived from leaves [12].

The current study is dedicated in its first part to the valorization of the roots of the olive tree by identifying the active phytochemical substances through the use of various chromatographic techniques. In its second part, our central focus is on the biological activities of the extracts.

## 2. Results and Discussion

### 2.1. The Quantitative Study of the Different Extracts

The extraction yield of dry residue rich in phenolic compounds from the external part of the olive roots was 31.16 g/100 g, the internal part was 10.23 g/100 g, and the root sum was 15.72 g/100 g. The percentages of extraction yields are illustrated in Table 1.

Our results show that extract yields for all three samples are important, ranging on average from 10.23% to 31.61%. The highest yield was registered for the outer part of the olive root, with a value of 31.61%.

The quantitative analysis of the different phenolic compounds was undertaken by high-performance liquid chromatography (HPLC). Several compounds from different families (flavonoids, secoiridoids, lignans, iridoids, organic acids, and sugars) were quantified (Table 2), such as oleuropein, which is the major compound, whose concentration varies between 5.892 mg/g for the root bark and 10.362 mg/g for the inner part of the root; elenolic acid, with concentrations varying from 4.54 mg/g to 0.116 mg/g; oleuropein hexoside content ranges from 3.484 mg/g to 0.141 mg/g;and ligstroside, which varied from 2.112 mg/g to 5.241 mg/g. Figure 1 displays the structure of the detected main compounds.

Figure 2 illustrates a comparison between the content of selected compounds in roots and dry leaves studied by Talhaoui et al. [13]. The oleuropein content in olive roots varies from 5.892 to 10.362 mg/g, while that in leaves is around 18.01 mg/g. The verbascoside concentration in dry leaves is around 4 mg/g, while in the inner part of olive roots it amounts to 3.207 mg/g. Moreover, the ligesroside content of olive roots (5.241 mg/g) is higher than that of olive leaves (3.845 mg/g).

### 2.2. Qualitative Profiling via HPLC-DAD and LC-ESIMS/MS and Their Relative MS/MS Data

The identified phenolic compounds marked by HPLC-DAD and LC-ESI- MS/MS in negative ionization mode are exhibited in Table 3. Each compound is related to its retention time (RT), its molecular monoisotopic neutral mass, its experimental *m*/*z*, its molecular formula, its mass error (in ppm), and its main MS/MS fragments.

To fully assess the phenolic composition of olive roots, we have chosen to study each part on its own in order to compare them with each other. The first part corresponds to the outer part or bark as represented by the chromatogram (A); the second part corresponds to the inner part of the root as represented by the chromatogram (B); and the third part corresponds to the sum of the roots as represented by the chromatogram (C). The base-peak chromatograms (BPC) of olive roots are plotted in Figure 3, indicating their richness in phenolic compounds, as 40 phenolic compounds were identified for the roots. Olive metabolites were classified into eight structural classes: sugars and derivatives, organic acids, phenolic acids (simple), flavonoids, caffeoyl phenylethanoid derivatives, iridoids and derivatives, secoiridoids and derivatives, and lignans.

#### 2.2.1. Sugars and Derivatives

Low-molecular-weight sugars share multiple chemical properties. They are optically active aliphatic polyhydroxy compounds that are generally highly soluble in water. The presence of sugars and their derivatives was denoted with a peak at 1.49 min, representing the monosaccharide D-mannitol (C_6_H_14_O_6_) (181.07 *m*/*z*) with its fragment (163.06 *m*/*z*) (C_6_H_12_O_5_), corresponding to the loss of water molecules (-H_2_O), (89.0244 *m*/*z*), (71.0141 *m*/*z*), and (119.0343 *m*/*z*). This is in good agreement with previous data addressing olive wood from Spanish cultivars [14,15].

#### 2.2.2. Organic Acids

Organic acids are ubiquitous in nature, as they are found in animal, vegetable, and microbial sources. They contain one or more carboxylic acid groups that can be covalently bonded with such groups as amides, esters, and peptides. Five compounds were observed in olive root extracts, including: gluconic acid (C_6_H_12_O_7_) (195.05 *m*/*z*) at 1.657 min with their two fragments (177.0418 *m*/*z*) (C_6_H_10_O_6_) and (159.0282 *m*/*z*) (C_6_H_8_O_5_) corresponding to a successive dehydration (-2H_2_O), and another fragment of molecular mass (129.0192 *m*/*z*) (C_5_H_5_O_4_) ascribed to the loss of the group (-CH_2_OH). Threonic acid (135.02 *m*/*z*) (C_4_H_8_O_5_) is represented by a peak at 1.87 min, with a fragment (117.0167 *m*/*z*) assigned to the elimination of a water molecule (H_2_O). Another fragment (89.0242 *m*/*z*) corresponded to the loss of the group (HCOOH) from threonic acid. Malic acid (C_4_H_6_O_5_) was identified at 1.95 min with (133.01 *m*/*z*) along with the main fragments: (115.0038 *m*/*z*) and (71.0142 *m*/*z*). Citric acid (191.0177 *m*/*z*) and 3,4-dihydroxyphenylacetic acid (167.03 *m*/*z*). The fragmentation pattern of the olive root extracts displayed similar results to those reported in the literature [16,17,18].

#### 2.2.3. Flavonoids

Luteolin (C_15_H_9_O_6_) (285.18 *m*/*z*) at 7.4 min with the main fragments: (201.0876 *m*/*z*), (270.1543 *m*/*z*), (255.1314 *m*/*z*), (131.9020 *m*/*z*), (114.9548 *m*/*z*), and (135.0262 *m*/*z*). The identification was carried out by comparing their ECs and fragmentation pathways to those reported in the literature and databases [18,19].

#### 2.2.4. Caffeoylphenylethanoid Derivatives

Verbascoside and isoverbascoside isomers were identified at (623.1983 *m*/*z*) (C_29_H_36_O_15_). The main fragment at (461.16 *m*/*z*) was obtained from the loss of the glucose (C_6_H_11_O_5_) residue from the deprotonated molecular ion (623.1983 *m*/*z*). Other fragments were present, such as (161.02 *m*/*z*), (113.02 *m*/*z*), (135.04 *m*/*z*), and (315.11 *m*/*z*). Their retention pattern, MS, and MS/MS spectra are in good accordance with the available data [10,13,18].

#### 2.2.5. Phenylethanoids

Two peaks are present at 4.38 and 4.66 min with precursor ion values equal to (315.1085 *m*/*z*) and (153.0546 *m*/*z*), corresponding to hydroxytyrosol-glucoside (C_14_H_20_O_8_) and hydroxytyrosol (C_8_H_10_O_3_), respectively. The MS-MS fragmentation spectrum of hydroxytyrosol exhibits a main fragment at (123.0447 *m*/*z*) obtained through the loss of the (CH_2_OH) group at(123.0447 *m*/*z*).

#### 2.2.6. Iridoids and Derivatives

Iridoids are known to offer a wide range of health benefits, including resisting a wide assortment of physical, biological, and chemical stressors. Multiple scientific studies have highlighted that iridoids possess antioxidant, antibacterial, anti-cancer, and anti-viral properties. Iridoids have been identified in olive tree root extracts. Among the most important ones, we state loganic acid (C_16_H_24_O_10_) (375.12 *m*/*z*) with main fragments at (213.0763 *m*/*z*), (341.1061 *m*/*z*); secologanosid (389.1087 *m*/*z*) (C_16_H_22_O_11_) (Figure 4); and main fragments at (345.1138 *m*/*z*), (183.0676 *m*/*z*), (121.0673 *m*/*z*), and (227.0571 *m*/*z*) related to the loss of (C_6_H_11_O_5_), (89.0225 *m*/*z*), and (165.0560 *m*/*z*). Relying on the spectrophotometric data, Figure 5 illustrates a proposed mechanism for the fragmentation of secologanosid. The appearing peak has a retention time of 19.35 min (359.13 *m*/*z*), and the fragments (197.0814 *m*/*z*), (153.0922 *m*/*z*), (135.0815 *m*/*z*), and (109.0668 *m*/*z*) have been identified as 7-deoxyloganic acid (C_16_H_24_O_9_). The peak with a retention time of 11.24 min (389.14 *m*/*z*) and the fragments (345.1559 *m*/*z*), (115.0395 *m*/*z*), (301.1637 *m*/*z*), (151.0773 *m*/*z*), (101.0252 *m*/*z*), and (83.0132 *m*/*z*) were identified as loganin.

This type of monoterpene is widely distributed in the Oleaceae (Obied et al., 2008) [20]. It is to be noted that their fragmentation patterns are consistent with previous works [10,14,21].

#### 2.2.7. Secoiridoids and Derivatives

Secoiridoids are the major class of secondary metabolites identified in olives. Among them, oleuropein corresponds to the major compound in leaf, stem, and root extracts. Several secoiridoids were detected in the extracts, and all were identified by comparing their ECs and fragmentation pathways to those found in the literature and databases [10,16].

A total of 22 known secoiridoids have been tentatively identified in olive roots. They notably correspond to oleoside (C_16_H_22_O_11_) (389.1087 *m*/*z*), acyclodihydroelenolic acid hexoside I (C_17_H_28_O_11_) (407, 1559 *m*/*z*), oleoside methyl ester derivative (C_18_H_30_O_11_) (421.1697 *m*/*z*),elenolic acid hexoside(C_17_H_24_O_11_) (403.1243 *m*/*z*) methyl oleuropeinaglycone (C_17_H_28_O_10_) (391.1602 *m*/*z*), 2″-ethoxyoleuropein (C_27_H_36_O_14_) (583.2032 *m*/*z*), hydroxyoleuropein (C_25_H_32_O_14_) (555.1759 *m*/*z*), demethyloleuropein (C_24_ H_30_ O_13_) (525.1618 *m*/*z*), oleuropein hexoside (C_31_H_42_O_18_) (701.2319 *m*/*z*),oleuropein hexoside (C_31_H_42_O_18_) (701.2304 *m*/*z*), methoxyoleuropein (C_26_H_34_O_14_) (569.498 *m*/*z*), dihydrooleuropein (543.2316 *m*/*z*), oleuropein (C_25_H_32_O_13_) (539.1781 *m*/*z*), lucidumoside C (C_27_H_36_O_14_) (583.1732 *m*/*z*), 60-O-[(2E)-2.6-dimethyl-8-hydroxy-2-octenoyloxy]-secologanoside (C_26_H_38_O_13_) (557.2225 *m*/*z*), ligustroside (C_25_H_32_O_12_) (523.1822 *m*/*z*), ligstrosideaglycone (C_19_H_22_O_7_) (361.1193 *m*/*z*), jaspolyoside (C_42_H_54_O_23_) (925.3006 *m*/*z*), hydroxyoleuropeinaglycone (C_19_H_22_O_9_) (393.1194 *m*/*z*) and oleuropein derivate (C_35_H_46_O_15_) (705.2774 *m*/*z*), elenolic acid hexoside derivative (C_19_H_26_O_13_) (461.1301 *m*/*z*) and nuezhenide (C_31_H_42_O_17_) (685.2371 *m*/*z*). We opted to characterize two compounds by examining their MS and MS/MS spectra. Oleuropein at 25.586 min is present at (539.1811 *m*/*z*) (C_25_H_31_O_12_). Two main fragments at (403.1217 *m*/*z*) and (377.1217 *m*/*z*) are obtained by the loss of the (C_8_H_9_O_2_) moiety as well as the loss of the glucose residue group of the deprotonated molecular ion (539.1811 *m*/*z*). Other characteristic fragments can be detected, such as (275.0898 *m*/*z*), which is characteristic of the loss of (C_4_H_6_O_3_) from (377.1217 *m*/*z*), and (307.0792 *m*/*z*), which is characteristic of the loss of (C_4_H_6_O) from (377.1217 *m*/*z*). Ligstroside at 28.42 min, another well-known compound of the Oleaceae family, is present with 523.1811 *m*/*z*(C_25_H_31_O_12_) (Figure 6). The main fragment (361.1277 *m*/*z*) is obtained by the loss of a glucose residue from the deprotonated molecular ion (523.1811 *m*/*z*). Other characteristic fragments can be observed, such as (291.0865 *m*/*z*), obtained through the loss of (C_4_H_6_O) from (361.1297 *m*/*z*), and (259.0975 *m*/*z*), characteristic of the loss of (CH_3_OH) from (291.0865 *m*/*z*). Figure 7 depicts a proposed mechanism for the fragmentation of ligstroside. The fragmentation model was checked through comparison with previous studies [22].

#### 2.2.8. Lignans

Another equally significant group of compounds present in the olive tree, namely the lignans, characterizes mainly the stem and the roots. After careful interpretation of their MS and MS/MS spectra, the identified compounds are: olivil (C_20_H_24_O_7_) (375.1476 *m*/*z*), and their main fragments are: (195.0672 *m*/*z*), (179.0725 *m*/*z*), (161.0561 *m*/*z*), and (122.0408 *m*/*z*). This is in good conformity with the opening and cleavage of the tetrahydrofuran ring, the additional loss of methyl groups, and the loss of water molecules, as reported by Sanz et al. [23]. Olivil 4-O-β-D-glucopyranoside (C_26_H_34_O_12_) was characterized by a peak appearing at 15.76 min with (537.1990 *m*/*z*) and the fragments: (375.1448 *m*/*z*), (179.0703 *m*/*z*), (195.0666 *m*/*z*), (345.1343 *m*/*z*), and (327.1202 *m*/*z*). Another peak emerged at 15.44 min with (415.1581 *m*/*z*) and the fragments: (149.0447 *m*/*z*), (89.0233 *m*/*z*), (191.0577 *m*/*z*), (251.0803 *m*/*z*), and (131.0396 *m*/*z*) were identified as 1-acetoxy-pinoresinol (C_19_H_28_O_10_). All aforementioned lignans were previously reported in olive wood [14,15].

Figure 8 illustrates the extracted ion chromatogram (EIC) profiles and MS-MS spectra of selected phenolic compounds identified in methanolic extracts from Chemlali olive roots.

### 2.3. Cytotoxicity and Antiviral Activity

The dose-response effect of the extracts (external part M-EX, internal part M-IN, and all the roots M-T) on the viability of VERO cells is portrayed in Figure 9. The exact CC_50_ values (concentrations of the tested extract decreasing the cellular viability by 50%) could not be assessed but were above 500 µg/mL for all tested extracts. According to a previously published work [24], no significant cytotoxicity of plant extract is recorded when the CC_50_ is above 500 µg/mL.

None of the tested extracts displayed any noticeable effect on the formation of virus-induced cytopathic effects(CPEs) in the infected VERO cells. Subsequent end-point dilution assays revealed that the infectious titers of both viruses in the extract-treated samples were comparable to the titers in the appropriate virus control samples.

At this stage of analysis, it is noteworthy that the extracts obtained from the underground parts of the olive tree Chemlalai cultivar exhibit no significant cytotoxicity towards non-cancerous VERO cells. Interestingly, methanolic extracts from the twigs of Chemlali and Chétoui cultivars presented moderate cytotoxicity towards VERO cells with CC_50_ values of 155 and 151 μg/mL [25]. Aqueous extracts from the leaves of two olive tree varieties (Meski and Chemlali) proved to increase the viability of murine oligodendrocytes (158N) in the concentration range of 5–200 μg/mL, while at 400 μg/mL, the viability of 158N was reduced [26]. The olive leaf ethanolic extracts also suppressed cytotoxicity induced by oxidative stress on the mouse embryonic fibroblast (MEF) [27]. Hydroethanolic extract from the leaves of the Chemlali cultivar proved to inhibit the proliferation of a cell line derived from human chronic myeloid leukemia (K562) and to exert G0/G1 cell cycle arrest on the 1st and 2nd days of incubation, and then also at G2/M phase (3rd and 4th days). Additionally, the extract induced apoptosis and differentiation of K562 cells towards the monocyte lineage [28]. Essafi Rhouma et al. [29] extracted the flowers of the Chemlali cultivar with 80% (*v*/*v*) ethanol and tested their cytotoxicity towards breast cancer cells (MCF-7) and human mammary epithelial cells (MCF-10A). The extracts in the concentration range between 31.25 and 250 μg/mL decreased the viability of MCF-7 cells without displaying a detrimental effect on MCF-10A. Moreover, the western blot analysis revealed the potential pro-apoptotic activity of flower extracts in MCF-7 cells [29].

None of the tested extracts demonstrated antiviral activity against the CVB3 or HHV-1 viruses replicating in the VERO cells. Leila et al. [25] investigated the cytotoxicity and antiviral activity against CVB3 or HHV-2 (Human Herpesvirus type 2) of twig extracts obtained using hexane, dichloromethane, ethyl acetate, and methanol from two Tunisian olive cultivars (Chemlali and Chétoui). Extracts from both cultivars displayed no effect on the replication of HHV-2. However, hexane extracts from Chemlali and Chétoui cultivars exhibited antiviral effects towards CVB-3 with IC_50_ (50% inhibitory concentration) of 20.41 and 23.28 μg/mL, respectively, and a selectivity index (SI) > 5 [25]. The hydroethanolic (water:ethanol; 50:50 (*v*/*v*)) extracts from the leaves of *O. europaea* var. *sativa* and *O. europaea* var. *sylvestris* were found to inhibit the replication of HHV-1 in the HeLa cells, reducing the infectious titer (plaque-forming assay) as well as the viral load (real-time PCR) [30]. Despite the literature data describing olive twigs or leaves as potential sources of antiviral compounds, we have not observed any antiviral activity for the extracts from underground parts of the olive tree (Chemlalai cultivar). In the case of hexane extracts from twigs of Chemlali and Chétoui cultivars, antiviral activity against CVB-3 was observed, but no detailed phytochemical analysis was performed. It was suggested that 2,4-di-tert-butylphenol was the bioactive compound responsible for this activity, but this was only shown through bio-guided assays using thin-layer chromatography [25]. This compound was not present in the root extracts we studied. The *O. europaea* var. *sativa* and *O. europaea* var. *sylvestris* extracts showed potent anti-HHV-1 activity, and this activity was probably due to the presence of oleuropein [30]. We have also identified oleuropein in the extracts from underground parts. Still, the content of this compound is the highest in leaves, and this may be the reason for the lack of antiviral activity of root extracts. Unfortunately, the cytotoxicity of root extracts did not allow us to test the antiviral activity using higher concentrations than those presented in Figure 10.

## 3. Materials and Methods

### 3.1. Sample Extraction

The olive root (2 kg) was collected from Chemlali olive trees grown in the Sfax region (Tunisia). The olive trees were about 45 years old. Olive root samples were collected in early January 2021. The samples were dried for 90 days at room temperature in a dark and airy room, and then the root was scraped at a local sawmill to separate the bark from the root wood. All samples were then ground to a fine powder using a domestic mill before being extracted. The roots of the olive tree have been separated into three parts, namely the external part, the internal part, and the total roots. Each part of the olive roots was placed separately in amber glass bottles and homogenized in a solution of methanol/water 80:20 (*v*/*v*) using an ultrasonic bath for 30 min. After filtration with Whatman filter paper No. 42 (125 mm), the extracts were put in a rotary evaporator under vacuum at 40 °C. Eventually, the extracts were kept at −20 °C until future analysis. After extraction and obtaining dry residues, the extraction yield was computed.

### 3.2. HPLC Analysis

In order to characterize the phenolic compounds by HPLC (Agilent 1200 series), the mobile phases relied on a combination of solvent A (water) and solvent B (methanol). The following multi-step gradient was applied: 0 min, 50% B; 5 min, 60% B; 20 min, 80% B; 25 min, 100% B; 26 min, 50% B; 35 min, 50% B. The flow rate was set at 1 mL/min throughout the gradient. The separation was undertaken with a Gemini 5 mcm NX-C18 110A column (250 × 4.6 mm) at room temperature. UV spectra were recorded from 190 to 600 nm. The injection volume was 10 uL. The identification of each compound was based on a comparison of their retention times with those of the reference compounds and recording the UV spectra in the 190–400 nm range. The quantitative determination was performed using calibration curves for phenolic standards. The content of the phenolic compounds was expressed in mg/g of dry olive roots.

### 3.3. MS-MS Analysis

The system was coupled to a 6540 Agilent Ultra-High-Definition Accurate-Mass QTOF, equipped with an ESI source (Agilent Dual Jet Stream) (Agilent Technologies, Palo Alto, CA, USA). The operating conditions in negative ionization mode were as follows: gas temperature, 350 °C; drying gas, nitrogen at 12 L/min; nebulizer pressure, 40 psig; sheath gas temperature, 400 °C; sheath gas flow, nitrogen at 12 L/min; capillary voltage, 4000 V; skimmer, 645 V; octopol radiofrequency voltage, 750 V, with the corresponding polarity automatically set. The spectra were acquired over a mass range from 100 to 1000 *m*/*z*, and for MS/MS experiments, from 100 to 1000 *m*/*z*. The reference mass correction of each sample was carried out with a continuous infusion of Agilent API TOF reference mixture (61,969–85,001). The data analysis was conducted on a Mass Hunter Qualitative Analysis B.10.00 (Agilent Technologies).

### 3.4. Cytotoxicity and Antiviral Activity

The VERO cells were cultured using Dulbecco’s Modified Eagle’s Medium (DMEM, Corning, Tewksbury, MA, USA) supplemented with fetal bovine serum (FBS; Capricorn Scientific, Ebsdorfergrund, Germany) and antibiotics (Penicillin-Streptomycin Solution, Corning). The dimethyl sulfoxide and 3-(4,5-dimethylthiazol-2-yl)-2,5-diphenyltetrazolium bromide (MTT) were obtained from Sigma-Aldrich (St. Louis, MO, USA), whereas phosphate-buffered saline (PBS) and trypsin were purchased from Corning. Incubation was undertaken at 37 °C in a 5% CO_2_ atmosphere (CO_2_ incubator, Panasonic Healthcare Co., Ltd., Tokyo, Japan).

HHV-1 (ATCC, No. VR-260) and CVB3 (ATCC, No. VR-30) were propagated in VERO cells, and the infectious titers (CCID_50—_50% cell culture infectious dose) were assessed using an end-point dilution assay.

#### 3.4.1. Cytotoxicity Testing

Cytotoxicity was tested according to the previously described methodology [31]. The monolayer of VERO cells in 96-well plates was treated with serial dilutions of extracts in cell media for 72 h. Afterwards, the media were removed. After washing with PBS, the MTT solution in DMEM was added, and the incubation continued for 4 h. Eventually, the formazan crystals were dissolved using SDS/DMF/PBS solvent. Following the overnight incubation, the absorbance (540 and 620 nm) was determined using the Synergy H1 Multi-Mode Microplate Reader (BioTek Instruments, Inc., Winooski, VT, USA). Collected data were exported from Gen5 software (ver. 3.09.07; BioTek Instruments, Inc., Winooski, VT, USA) to GraphPad Prism (v7.0.4) for further analysis.

#### 3.4.2. Antiviral Activity

The antiviral activity was tested, as previously described [31]. The VERO cells seeded in 48-well plates were infected with HHV-1 or CVB3 at a 100-fold CCID_50_ infectious titer. After 1 h of pre-incubation allowing virus attachment, the cell monolayer was washed with PBS and treated with extracts in non-toxic concentrations. The non-toxic doses were selected based on dose-response curves obtained during cytotoxicity studies. They were defined as concentrations of selected extracts that did not reduce cellular viability by more than 10%. Incubation continued until a typical CPE was observed in the virus control (infected, non-treated cells). Next, the sample-treated infected cells were also observed for the occurrence of CPE using an inverted microscope (CKX41, Olympus Corporation, Tokyo, Japan) equipped with a camera (Moticam 3+, Motic, Hong Kong), and the results were recorded (Motic Images Plus 2.0, Motic, Hong Kong). Afterwards, the plates were thrice frozen (−72 °C) and thawed, and the samples were collected and kept frozen at −72 °C until the assessment of the viral infectious titer was carried out.

Virus titrations were conducted using an end-point dilution assay [31]. Briefly, monolayers of VERO in 96-well plates were treated with tenfold dilutions of samples from the above-described antiviral assays. After 72 h, the viral titers were calculated using an MTT assay. The antiviral activity was specified, referring to the difference in viral titer between the extract-treated samples and the virus control. A significant antiviral activity needs at least a 3 log reduction of the viral infectious titer.

## 4. Conclusions

Overall, olive roots may be considered a source of polyphenolic compounds. Spectrometric analysis applied through the use of the liquid chromatography-mass spectrometry technique proved to be a powerful tool for the characterization of several compounds in olive roots. 41 molecules of different families were identified, including secoiridoids, iridoids, lignans, and organic acids. Moreover, according to the quantification by UHPLC-DAD, oleuropein is the major compound detected in olive roots, with a concentration of 10.36 mg/g. Ligestroside has also been detected, and its amount is higher than that in leaves. These results provide a deeper and better insight into the bioactive compounds in the less-considered organs of the olive tree. Several compounds remained unidentified, revealing the chemical diversity of Olea. The extracts proved to be non-toxic to VERO cells and did not exert any noticeable antiviral effect against HHV-1 or CVB3 replicating in VERO cells. Indeed, in future works, we may tackle the identified compounds, attempt to isolate and investigate their characteristics, and fully understand how significant they are.

## Figures and Tables

**Figure 1 molecules-28-04829-f001:**
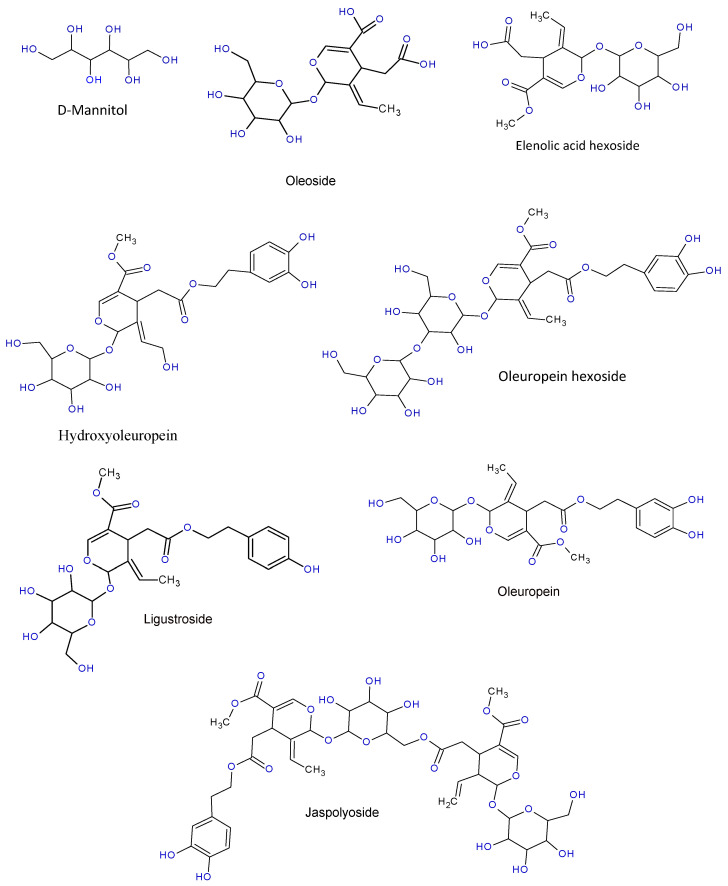
Structures of the major quantified compounds.

**Figure 2 molecules-28-04829-f002:**
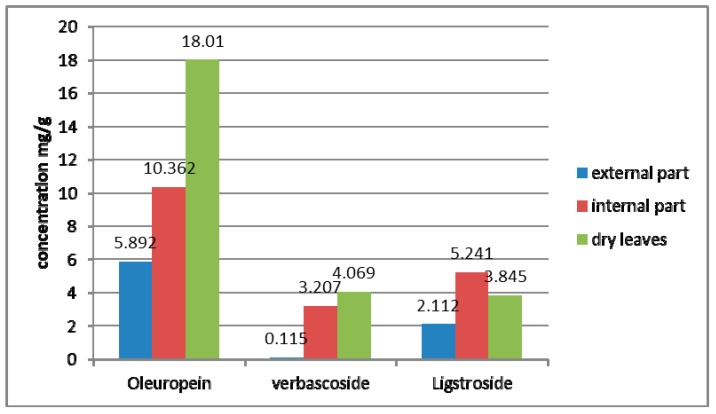
Comparison of the content of selected compounds in roots and leaves.

**Figure 3 molecules-28-04829-f003:**
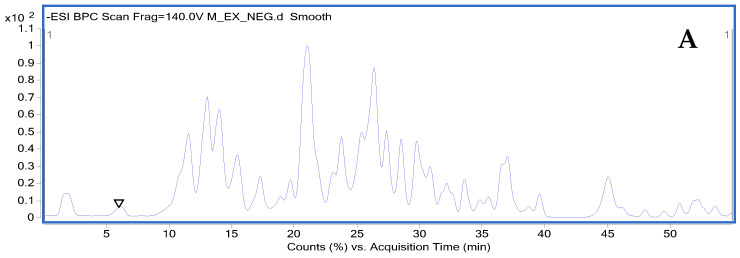
Base peak chromatogram (BPC) of the olive root methanol extract obtained by HPLC-DAD-ESI-MS in the negative ionization mode ((**A**)—chromatogram of the extract from the outer part of the olive tree root; (**B**)—chromatogram of the extract from the inner part of the olive tree root; (**C**)—chromatogram of the extract from the olive tree whole roots).

**Figure 4 molecules-28-04829-f004:**
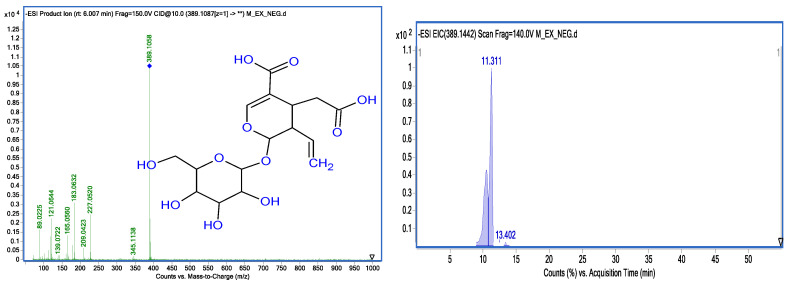
Ion chromatogram and MS-MS spectra of secologanosid.

**Figure 5 molecules-28-04829-f005:**
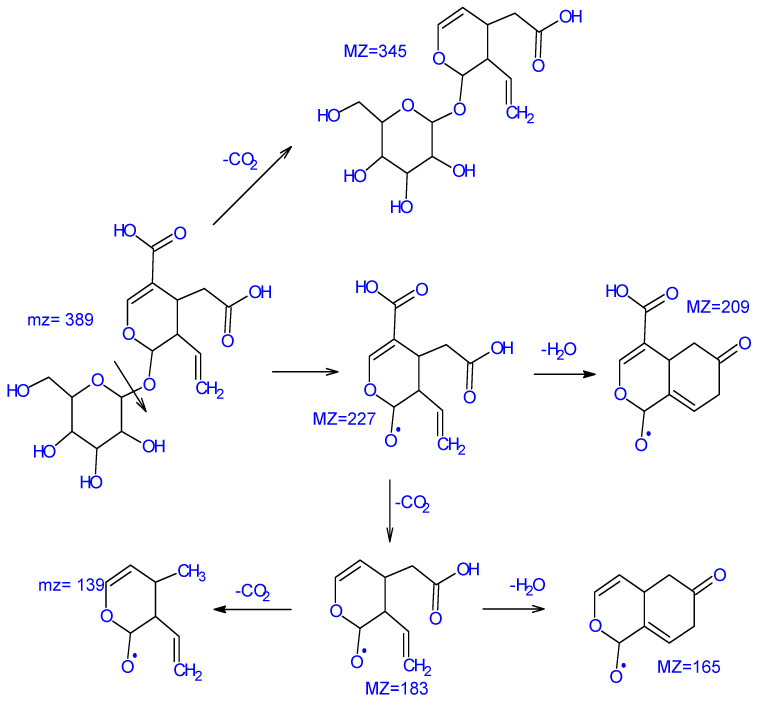
Proposed mechanism of secologanosid fragmentation based on MS/MS data.

**Figure 6 molecules-28-04829-f006:**
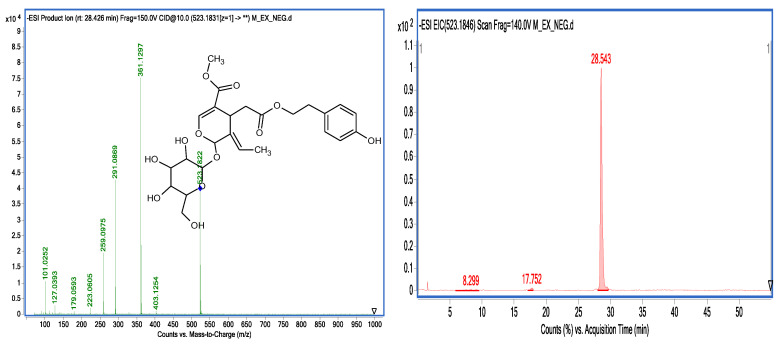
Ion chromatogram and MS-MS spectra of ligstroside.

**Figure 7 molecules-28-04829-f007:**
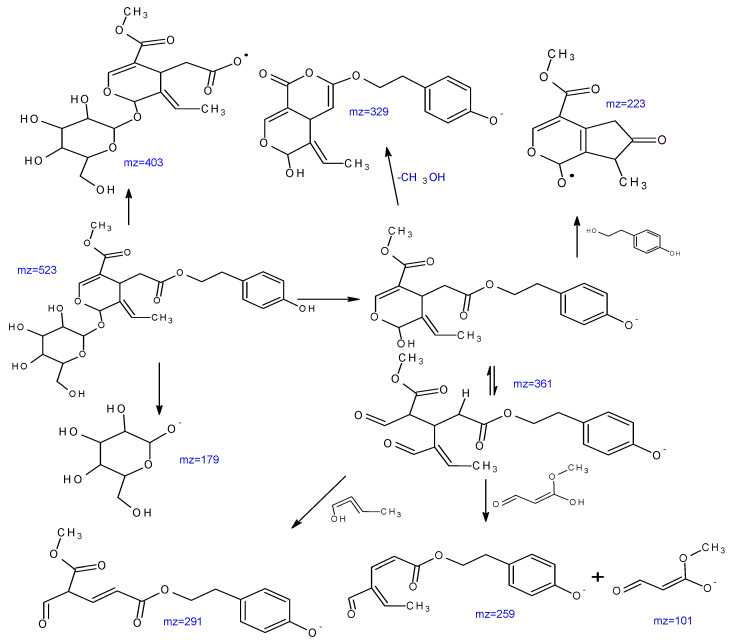
Proposed mechanism of ligestrosid fragmentation based on MS/MS data.

**Figure 8 molecules-28-04829-f008:**
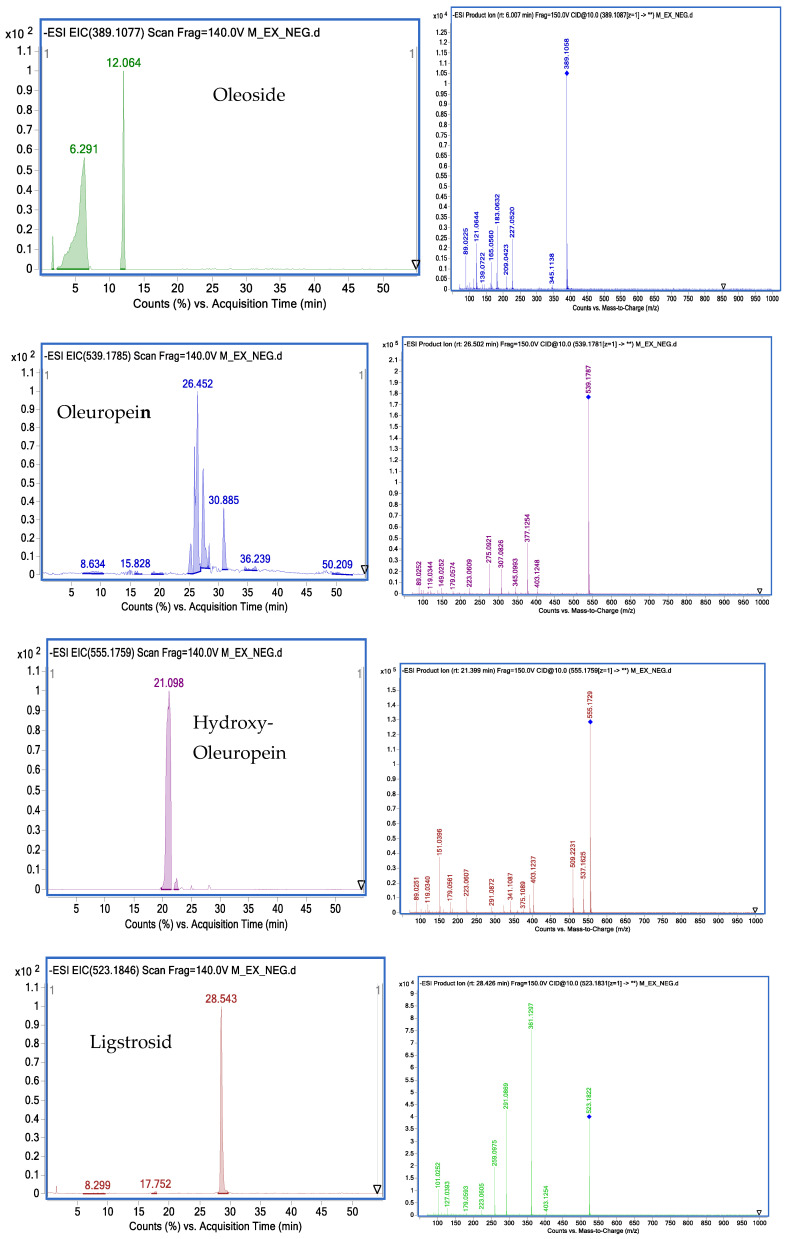
Profiles of extracted ion chromatograms (EIC) and their MS-MS spectra obtained by HPLC-DAD-MS of some phenolic compounds identified in methanolic extracts of Chemlali olive tree roots.

**Figure 9 molecules-28-04829-f009:**
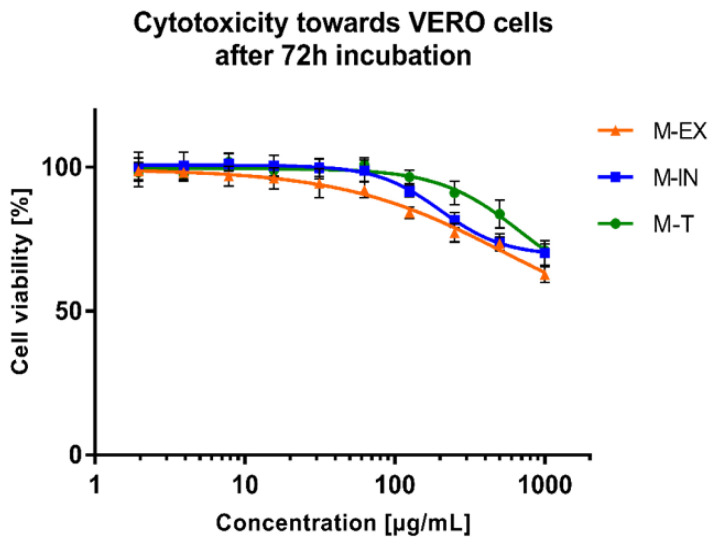
The cytotoxicity of extracts. (M-EX—external part; M-IN—internal part; M-T—whole roots).

**Figure 10 molecules-28-04829-f010:**
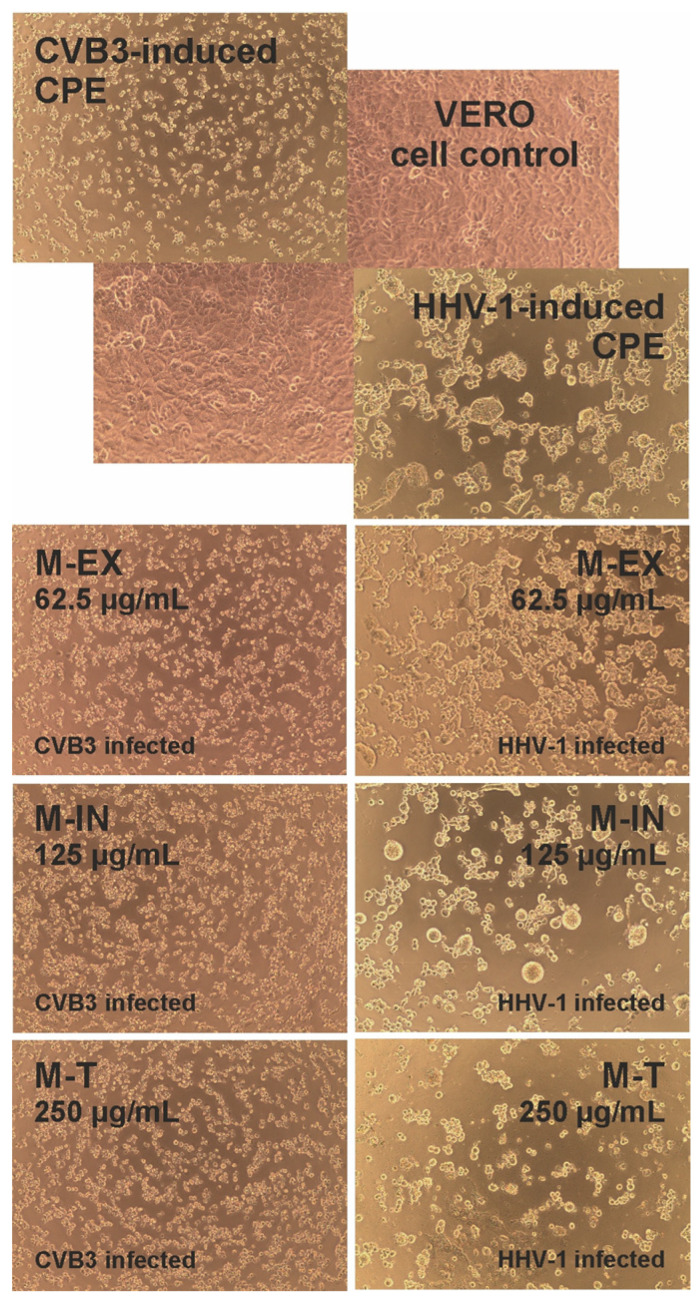
The influence of extracts on viral-induced cytopathic effects (CVB3—Coxsackievirus B3; CPE—cytopathic effect; HHV-1—Human Herpesvirus type 1; M-EX—external part; M-IN—internal part; M-T—whole roots).

**Table 1 molecules-28-04829-t001:** The extraction yield of the dry residue rich in phenolic compounds from olive tree roots.

Organs	Yield (%)
External part	31.61
Internal part	10.23
Root sum	15.72

**Table 2 molecules-28-04829-t002:** Content of the major detected compound.

N°	*m*/*z*	Compound	Concentration [mg/g]	
			External Part	Internal Part
1	181	D-Mannitol	0.493 ± 0.02	1.530 ± 0.06
2	389	Oleoside	0.317 ± 0.01	-
3	403	Elenolic acid hexoside	4.54 ± 0.19	0.116 ± 0.01
4	583	2″-Ethoxyoleuropein	2.227 ± 0.10	3.356 ± 0.13
5	555	Hydroxyoleuropein	4.892 ± 0.19	3.342 ± 0.15
6	701	Oleuropein hexoside	3.484 ± 0.14	0.141 ± 0.01
8	539	Oleuropein	5.892 ± 0.41	10.362 ± 0.44
9	623	verbascoside	0.115 ± 0.01	3.207 ± 0.13
10	523	Ligstroside	2.112 ± 0.09	5.241 ± 0.22
11	925	Jaspolyoside	2.648 ± 0.12	0.460 ± 0.018

**Table 3 molecules-28-04829-t003:** Identified compounds in olive roots using the negative ionization mode.

N°	RT (min)	Molecular Mass	[M-H]-	Molecular Formula	Mass Error (ppm)	Main Fragments via MS/MS	Proposed Compound	External Part	Internal Part	Total of the Root
sugar and derivates										
1	1.49	182.079	181.0708	C_6_H_14_O_6_	5.28	89.0244, 71.0141, 163.0600, 119.0343	D-Mannitol	+	+	+
organic acids										
2	1.657	196.0583	195.0500	C_6_H_12_O_7_	−5.23	177.0418, 159.0282, 75.0095, 129.0192, 99.0107	Gluconic acid	+	+	+
3	1.875	136.0372	135.0299	C_4_H_8_O_5_	−5.86	75.0095, 117.0167, 89.0242	Threonic acid	+	-	+
4	1.958	134.0215	133.0142	C_4_H_6_O_5_	−5.57	115.0038, 71.0142	Malic acid	+	+	+
5	2.410	192.027	191.0177	C_6_H_8_O_7_	0.55	111.0085, 87.0084, 129.0197 173.0060	Citric acid	+	+	+
6	5.221	168.0423	167.0354	C_8_H_8_O_4_	−6.44	123.0429, 149.0199, 109.0304 93.0356	3,4-dihydroxyphenylacetic acid	+	-	+
iridoids										
7	3.748	376.1369	375.1285	C_16_H_24_O_10_	1.41	213.0763, 341.1061	Loganic acid	+	+	+
8	6.007	390.1162	389.1087	C_16_H_22_O_11_	0.68	345.1138, 183.0676, 121.0673 227.0571, 89.0225, 165.0560	Secologanoside	+	+	+
9	11.244	390.1526	389.1442	C_17_H_26_O_10_	1.74	345.1559, 115.0395, 301.1637 151.0773, 101.0252, 83.0132	Loganin	+	-	+
10	19.358	360.142	359.1344	C_16_H_24_O_9_	−0.99	197.0814, 153.0922, 135.0815, 109.0668	7-Deoxyloganic acid	+	+	+
Phenylethanoids										
11	4.384	316.1158	315.1085	C_14_H_20_O_8_	−3.93	153.0539, 135.0447, 101.0248	Hydroxytyrosol-glucoside	+	+	+
12	4.669	154.063	153.0546	C_8_H_10_O_3_	−7.26	123.0447	Hydroxytyrosol	+	+	+
Caffeoyl phenylethanoid derivatives										
13	23.055	624.2054	623.1983	C_29_H_36_O_15_	−1.19	461.1662, 161.0238, 113.0242 135.0451, 315.1103	verbascoside	+	+	+
14	23.089	624.2054	623.1983	C_29_H_36_O_15_	−1.19	461.1662, 161.0238, 113.0242 135.0451, 315.1103	Isoverbascoside	+	+	+
Phenolic acids										
15	7.814	448.1581	447.1508	C_19_ H_28_ O_12_	3.06	153.0547, 285.0980, 363.1466 89.0221, 112.9829	Dihydroxybenzoic acid hexoside pentoside	+	-	+
Flavonoids										
16	46.227	286.1846	285.1866	C_15_H_10_O_6_	2.18	201.0876, 270.1543, 255.1314, 131.9020, 114.9548, 135.0262	Luteolin	+	-	+
Secoiridoids and derivatives										
17	4.970	390.1162	389.1087	C_16_H_22_O_11_	0.6	183.0681, 227.0553, 121.0674 209.0414, 165.0562	Oleoside	+	-	+
18	6.559	408.1632	407.1559	C_17_H_28_O_11_	−3.88	389.1447, 357.1187, 89.0239,	Acyclodihydroelenolic acid hexoside I	+	-	+
19	9.738	422.1788	421.1697	C_18_H_30_O_11_	−0.79	403.1614, 359.1688, 115.0389, 101.0235, 73.0283, 379.1549 151.0737	Oleoside methyl ester derivative	+	+	+
20	11.528	404.1319	403.1243	C_17_H_24_O_11_	−0.71	89.0251, 223.0607, 119.0338, 179.0564, 71.0146, 101.0236	Elenolic acid hexoside	+	+	+
21	13.837	392.1682	391.1602	C_17_H_28_O_10_	−0.18	225.1114, 183.1017, 255.1255, 285.1284, 167.0704, 113.0238, 89.0228	Methyl oleuropein aglycone	+	+	+
22	14.038	584.2105	583.2032	C_27_H_36_O_14_	1.15	537.1984, 375.1453, 179.0710, 195.0667, 345.1339	2″-Ethoxyoleuropein	+	+	+
23	18.187	462.1373	461.1301	C_19_H_26_O_13_	0.08	403.1209,75.0099, 223.0617 149.0428	Elenolic acid hexoside Derivative	+	+	+
24	21.399	556.1792	555.1759	C_25_H_32_O_14_	1.75	151.0396, 509.2231, 537.1625 403.1237, 223.0607	Hydroxyoleuropein	+	+	+
25	21.282	526.1686	525.1618	C_24_H_30_O_13_	0.83	195.0662, 389.1059, 345.0980 319.1235, 209.0476	Demethyloleuropein	+	+	+
26	21.449	702.2371	701.2319	C_31_H_42_O_18_	2.65	315.1071, 539.1808, 469.1349 437.1439	Oleuropein hexoside	+	+	+
27	21.7	702.2371	701.2304	C_31_H_42_O_18_	0.8	315.1054, 539.1741, 469.1353 437.1428	Oleuropein hexoside	+	+	+
28	23.708	686.2422	685.2371	C_31_H_42_O_17_	−1.05	523.1801, 453.1276, 421.1484 299.1130	Nuezhenide	+	-	+
29	25.498	570.1949	569.498	C_26_H_34_O_14_	0.74	537.1619, 403.1237, 223.0612 151.0396, 179.0555, 119.0350 89.0250, 337.0904, 305.1026	Methoxyoleuropein	+	+	+
30	26.358	544.2215	543.2316	C_25_H_36_O_13_	1.18	377.1364, 197.0780, 153.0949 183.1068, 109.1033	Dihydro oleuropein	+	+	+
31	26.586	540.1843	539.1781	C_25_H_32_O_13_	2.01	403.1217, 377.1217, 307.0792 275.0898, 223.0594, 179.0533 149.0228, 119.0331, 89.0240	Oleuropein	+	+	+
32	27.054	584.1741	583.1732	C_27_H_36_O_14_	1.29	537.1590, 403.1224, 223.0573 151.0403, 179.0521	Lucidumoside C	+	-	+
33	28.058	558.2312	557.2225	C_26_H_38_O_13_	−2.62	513.2315, 227.1269, 371.0920 121.0659, 183.0636, 165.0614	60-O-[(2E)-2.6-Dimethyl-8- hydroxy-2-octenoyloxy]- secologanoside	+	-	+
34	28.426	524.1894	523.1822	C_25_H_32_O_12_	0.19	361.1297, 291.0869, 259.0975 101.0252, 127.0393, 179.0593	Ligustroside	+	+	+
35	28.644	362.1366	361.1193	C_19_H_22_O_7_	1.17	291.0882, 259.0991, 127.0401 101.0245, 171.0286	Ligstroside aglycone	+	+	+
36	29.580	926.3056	925.3006	C_42_H_54_O_23_	2.47	789.2470, 745.2361, 661.2086 539.1780, 521.1652, 403.1245 377.1234, 307.0824, 275.0915 223.0613, 149.0242, 119.0339 89.0247	Jaspolyoside I	+	+	+
37	31.354	394.1264	393.1194	C_19_H_22_O_9_	0.75	317.1027, 349.1270, 289.1046 181.0490, 137.0590, 153.0540	Hydroxy oleuropein aglycone	+	+	+
38	32,19	706.2837	705.2774	C_35_H_46_O_15_	1.42	539.1748, 521.1635, 377.1211 327.2138, 307.0799, 275.0896 223.0596, 149.0223	Oleuropein derivate	+	+	+
Lignan										
39	15.376	538.205	537.1990	C_26_H_34_O_12_	0.84	375.1448, 179.0703, 195.0666 345.1343, 327.1202	Olivil 4-O-β-D-glucopyranoside	+	+	+
40	15.510	376.1522	375.1476	C_20_H_24_O_7_	2.85	360.1440, 345.1361, 327.1244 195.0672, 179.0725, 161.0561 146.0376, 122.0408	Olivil	+	+	+
41	16.096	416.1682	415.1581	C_19_H_28_O_10_	−0.17	149.0447, 89.0233, 191.0577 251.0803, 131.0396	1-acetoxy-pinoresinol	+	-	+

## Data Availability

The data presented in this study are available on request from the corresponding author.

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
