# Peer review of "Comprehensive Metabolite Profiling of Chemlali Olive Tree Root Extracts Using LC-ESI-QTOF-MS/MS, Their Cytotoxicity, and Antiviral Assessment"

_molecules, 2023, doi:10.3390/molecules28124829_

Round 1

Reviewer 1 Report

The manuscript molecules-2421819 titled “Comprehensive Metabolite Profiling of Chemlali olive tree Roots. Extracts Using Liquid Chromatography Coupled with Electrospray Ionization Ion Mobility Quadrupole Time-of-Flight Mass Spectrometry and its biological activities” showed the identification of  phytochemical compounds from olive roots Olea europaea Chemlali cultivar using LC-MS technique. The crude methanolic extracts were evaluated for cytotoxicity against VERO cells using MTT assay.

The subject studied and the results obtained are considered preliminary for publishing extensive modification are required for publication in Molecules.

-        The title of the paper needs to be improved and must be shortened also the aim of work should be ameliorated.

-        Voucher specimen numbers of olive species must be included in the manuscript.

-        For the characterization of the olive root extracts:  the choice of the markers must be justified and quantification for the content of marker compounds. Hyopthesis generated that phenolics present in olive roots are very ubiquitous in many plant species.

-        The material and method part 1.2. HPLC analysis: The cited reference (Talhaoui et al. 2014) used another method for HPLC.  The analytical methods for quantifying the markers have to be "validated" and briefly described providing the most important information necessary to obtain reproducible results. 

-        Please clarify and please highlight the novelty of this study. What is the conclusion of this study?

-        The abstract should be ameliorated. The importance and main findings of this study are not clear. What was the impact of this study?

-        Table 3: what do you mean by TR and Erreur

-        Results of standard positive control are required for cytotoxicity assay.

-        None of the tested extracts demonstrated antiviral activity against the CVB3 or  HHV-1 viruses replicating in the VERO cells what is the aim of this experiment

-        Please separate the results and discussion and better describe the results.

-        The discussion merely states that this " this goes in good agreement with previous data", begging the question - have the authors merely replicated an already known phenomenon?

-        Please, the authors could speculate the weaknesses and strengths of their study in Discussion section.

-        I think the proposed mechanism of fragmentation based on MS/MS data is already published please highlight the new proposed information.

-        There are many typing mistakes in the manuscript and many minor corrections.

-        At many places genus and species names are not italicized. Correct these.

-        Please subscribe the number in the molecular formula of compounds example table 3 . Revise along the manuscript.

-        Please add the conclusion section

Author Response

According to reviewer #1:

The title of the paper needs to be improved and must be shortened also the aim of work should be ameliorated.

Answer: Dear Reviewer, thank you for this remark, we have changed the title “Comprehensive Metabolite Profiling of Chemlali olive tree Roots. Extracts Using Liquid Chromatography Coupled with Electrospray Ionization Ion Mobility Quadrupole Time-of-Flight Mass Spectrometry and its biological activities” to “Comprehensive metabolite profiling of Chemlali olive tree root extracts using LC-ESI-QTOF-MS/MS, their cytotoxicity and antiviral assessment.” also As the review suggests, we have ameliorate the aim of our work.

Voucher specimen numbers of olive species must be included in the manuscript.

Answer: As the reviewer suggests, we have added how we collected the raw material in the "sample extraction" section.

For the characterization of the olive root extracts:  the choice of the markers must be justified and quantification for the content of marker compounds. Hyopthesis generated that phenolics present in olive roots are very ubiquitous in many plant species.

Answer: Dear reviewer, the compound that could be used as a marker is oleuropein, as it was detected in both extracts (the extract from the outer part of the root and the extract from the inner part). Its concentration varied from 5.892mg/g for the external extract to 10.362mg/g for the internal extract.

The material and method part 1.2. HPLC analysis: The cited reference (Talhaoui et al. 2014) used another method for HPLC.  The analytical methods for quantifying the markers have to be "validated" and briefly described providing the most important information necessary to obtain reproducible results.

Answer: According to the reviewer comment, I agree with you. The method of HPLC analysis was revised and now is clearer for the reader.

Please clarify and please highlight the novelty of this study. What is the conclusion of this study?

Answer: Dear Reviewer, In future, when rejuvenating olive trees, by removing aged trees that don't give fruit; the roots will be exploited as they contain several phenolic compounds.

The abstract should be ameliorated. The importance and main findings of this study are not clear. What was the impact of this study?

Answer: Dear Reviewer, the impact of our study is to use olive tree roots to extract high value-added phenolic molecules.

Table 3: what do you mean by TR and Erreur

Answer: Dear Reviewer, thank you for this remark, TR: Retention time is used to identify and quantify compounds present in a sample, it is expressed in minutes.

The error refers to the difference between the real mass of an ion and the mass measured by the mass spectrometer. This difference is expressed in ppm.

Results of standard positive control are required for cytotoxicity assay.

Answer: Dear Reviewer, we understand your concern. However, there are no approved substances for negative or positive control of cytotoxicity towards non-cancerous cells. For our studies, we use cells supplemented with complete cell media as a negative control (cell control). We have not used a positive control because we were conducting a screening of plant extract activity towards non-cancerous VERO cells. Based on the published literature [Zengin et al. 2020; Łaska et al. 2019; Geran et al. 1972] no significant cytotoxicity of plant extract is recorded when the CC50 is above 500 µg/mL, and we have based the assessment of Olea europaea Chemlali cultivar extracts on those literature sources.

Various studies use cytotoxic compounds or anticancer drugs, like cisplatin, doxorubicin, etoposide, or hydroxycarbamide, as a positive control during in vitro studies of cytotoxicity towards cancer cells and anticancer properties. However, we believe that using those substances as a positive control in the studies of plant extract is not always necessary. Plant extracts are complex mixtures of various molecules, and it’s difficult to compare their activity with pure compounds. Of course, if particular compounds are isolated from plant extract and show significant anticancer potential, their activity should be compared to standard anticancer drugs.

  1. Zengin et al., “Chemical Characterization and Bioactive Properties of Different Extracts from Fibigiaclypeata, an Unexplored Plant Food,” vol. 9, p. 705, 2020, doi: 10.3390/foods9060705

Łaska, G., Sieniawska, E., Świątek, Ł., Zjawiony, J., Khan, S., Boguszewska, A., Stocki, M., Angielczyk, M., & Polz-Dacewicz, M. (2019). Phytochemistry and biological activities of Polemonium caeruleum L. Phytochemistry Letters, 30, 314-323. https://doi.org/10.1016/j.phytol.2019.02.017

Geran RI, Greenberg NH, Macdonald MM, Shumacher AM, Abbott BJ. Protocols for screening chemical agents and natural products against animal tumors and other biological systems. Cancer Chemotherapy Reports, Part III, 1972; 3: 1-103.

None of the tested extracts demonstrated antiviral activity against the CVB3 or HHV-1 viruses replicating in the VERO cells what is the aim of this experiment

Answer: Dear Reviewer, thank you for this remark. As we have mentioned in the manuscript, hydroethanolic extracts from the leaves of O. europaea var. sativa and O. europaea var. sylvestris were found to inhibit the replication of HHV-1 [Pennisi et al., 2023]. Also, twig extracts from Chemlali and Chétoui cultivars exhibited antiviral effect towards CVB-3 with IC50 (50% inhibitory concentration) of 20.41 and 23.28 μg/mL, respectively, and a selectivity index (SI) > 5 [Leila et al. 2019]. However, there are no reports on the antiviral activity of underground parts of olive tree. Thus we have undertaken an effort to fulfil this gap, and herein we report that no antiviral activity against HHV-1 nor CVB3 was observed.

  1. Pennisi et al., “Analysis of Antioxidant and Antiviral Effects of Olive (Olea europaea L.)leaf Extracts and Pure Compound Using Cancer Cell Model,” Biomolecules, vol. 13, no. 2, p. 238, Jan. 2023, doi: 10.3390/BIOM13020238
  2. Leila et al., “Isolation of an antiviral compound from Tunisian olive twig cultivars,” Microb. Pathog., vol. 128, pp. 245–249, Mar. 2019, doi: 10.1016/J.MICPATH.2019.01.012.

Please separate the results and discussion and better describe the results.

Answer: Dear Reviewer, for more clarity to the reader we have present result and discussion together, indeed after each result a discussion was presented to compare our result with those described previously.

The discussion merely states that this "this goes in good agreement with previous data", begging the question - have the authors merely replicated an already known phenomenon?

Answer: Dear Reviewer, no we didn't replicated an already phenomenon but we have compare our result with them and to find it similar.

Please, the authors could speculate the weaknesses and strengths of their study in Discussion section.

Answer: as requested by the reviewer, the discussion section was revised and ameliorated

I think the proposed mechanism of fragmentation based on MS/MS data is already published please highlight the new proposed information.

Answer: Dear reviewer, the LC-MS/MS fragmentation gives many peaks and in the proposed mechanism new fragment has been explained and was not yet published such as fragment (m/z 523) give fragment (m/z 403), fragment (m/z 361) give fragment (m/z 329) and (m/z 223).

There are many typing mistakes in the manuscript and many minor corrections.

Answer: As the reviewer suggests, we have reread the whole manuscript and corrected these errors. For English mistakes, an English teacher specializing in translation and language and review the whole manuscript

At many places genus and species names are not italicized. Correct these.

Answer: As the reviewer suggests, we have reread the whole manuscript and corrected them.

Please subscribe the number in the molecular formula of compounds example table 3. Revise along the manuscript.

Answer: As the reviewer suggests, we have corrected.

Please add the conclusion section

Answer: Dear Reviewer, the conclusion section was revised and ameliorated

Reviewer 2 Report

Plants are an important source of medicinal compounds. Olive trees have been used to produce edible oil since the antique history, and looking for bioactive molecules seems an interesting idea.

Indeed, there are already many publications about the extraction and biological evaluation of olive oils.

The authors should explain why they use tree roots in their study. And as well if their results justify the extraction of compounds from tree roots since the raw material cannot be harvested without seriously affecting the tree.

Is any relevant compound in higher concentration, compared to other regenerative parts, of the plant, such as leaves and fruits (experimental data)?

In my opinion, the current manuscript does not provide sufficient new information to justify publication. 

For example, in the conclusion, the authors state that LC-MS is a useful method for the analysis of olives, give some chemical and biological data, and suggests more research. But is there really any conclusion?

Specific comments:

Figure 2: What is A, B, C? The y-axis unit is uncommon/ odd for presenting a BPC.

Compound identification: Which are your acceptance criteria?

Table: In negative mode, it should be [M-H]-; Erreur = mass error?; ()-Olivil = ?

References: http://www.agriculture.tn/ is not a valid web reference. Besides, during review, this web page did not work.

The paper has several language problems and is too 'flowery' for a scientific paper. 

Example: the last part of the conclusions.

"As a final note, it is worth noting that although the obtained findings can be regarded as valuable and promising, this research work remains a step that can be extended and built upon as it offers further fruitful lines of investigation and opens future research directions. Indeed, in future works, we may tackle the unidentified compounds and attempt to decipher their characteristics and fully understand how significant they are."

Author Response

 According to reviewer #2:

The authors should explain why they use tree roots in their study. And as well if their results justify the extraction of compounds from tree roots since the raw material cannot be harvested without seriously affecting the tree.
Answer: Dear Reviewer, thank you for reviewing our manuscript. Since in our country aged olive trees that don't bear fruit are uprooted and replaced by young plants, the aim of this work is to make the most of the olive tree's roots, which are present in large quantities, and scientifically we want to see the difference between the composition of the roots and the stems of the olive tree.

Is any relevant compound in higher concentration, compared to other regenerative parts, of the plant, such as leaves and fruits (experimental data)?

Answer: Dear Reviewer, thank you for this question, the discussion section has been improved and a comparison made between the concentration of certain molecules in olive roots and leaves. In addition, it was found that the quantity of ligestroside in roots is 5.24 mg/g higher than in leaves 3,845mg/g which was studied by talhaoui et al.

  1. Talhaoui, A. M. Gómez-Caravaca, L. León, R. De la Rosa, A. Segura-Carretero, and A. Fernández-Gutiérrez, “Determination of phenolic compounds of ‘Sikitita’ olive leaves by HPLC-DAD-TOF-MS. Comparison with its parents ‘Arbequina’ and ‘Picual’ olive leaves,” LWT, vol. 58, no. 1, pp. 28–34, 2014, doi: 10.1016/j.lwt.2014.03.014.

In my opinion, the current manuscript does not provide sufficient new information to justify publication.

Answer:

Our country is the 3rd largest producer of olive oil in the world, and the olive grove consists of 105 million trees, 40% of which are old trees. The state is in the process of renewing these trees with young olive trees, which explains the large biomass of olive roots. This gives us reason to think that the residue, after extraction of the phenolic compounds, will be used for other treatments. This study was carried out for the first time on Tunisian olive varieties.

For example, in the conclusion, the authors state that LC-MS is a useful method for the analysis of olives, give some chemical and biological data, and suggests more research. But is there really any conclusion

Answer: Dear reviewer, the LC-MS method is an analytical technique widely used in many fields, such as chemistry, pharmacology and biology. It was used in our study for the identification and quantification of various olive root compounds

Figure 2: What is A, B, C? The y-axis unit is uncommon/ odd for presenting a BPC.

Answer: Dear Reviewer, thank you for this remark, we have chosen to study each part of the root separately; we have divided it into three parts, the outer part, the inner part and the sum. Each part is represented by a chromatogram. For the y-axis of the BCP chromatogram, we all changed the same axis.

Compound identification: Which are your acceptance criteria

Answer: Dear Reviewer, thank you for this question, the mass spectra obtained are compared with databases containing mass spectra of known compounds, this comparison enables compounds present in the sample to be identified on the basis of their characteristic mass spectra.

  1. Michel et al., “UHPLC-DAD-FLD and UHPLC-HRMS/MS based metabolic profiling and characterization of different Olea europaea organs of Koroneiki and Chetoui varieties,” Phytochem. Lett., vol. 11, pp. 424–439, 2015, doi: 10.1016/j.phytol.2014.12.020.
  2. Ammar, M. del M. Contreras, B. Gargouri, A. Segura-Carretero, and M. Bouaziz, “RP-HPLC-DAD-ESI-QTOF-MS based metabolic profiling of the potential Olea europaea by-product ‘wood’ and its comparison with leaf counterpart,” Phytochem. Anal., vol. 28, no. 3, pp. 217–229, 2017, doi: 10.1002/pca.2664.

Table: In negative mode, it should be [M-H]-; Erreur = mass error?; ()-Olivil = ?

Answer: As the reviewer suggests, these are typos and have all been corrected, we have changed "Erreur" to "mass error" and "()-Olivil" to "Olivil"

References: http://www.agriculture.tn/ is not a valid web reference. Besides, during review, this web page did not work.

Answer: As the reviewer suggests, we have corrected

The paper has several language problems and is too 'flowery' for a scientific paper.

Answer: as requested, we attempted to delete the “flowery language” incorporated in the conclusion section. An English teacher who is specialized in translation and language and review the whole manuscript.

Reviewer 3 Report

This work is devoted to the valorization of olive roots through identifying active phytochemicals and assessing their biological activities, including cytotoxicity and antiviral potential of different extracts from Olea europaea Chemlali cultivar. These studies bring a lot of new information about the phytochemistry of olive tree roots. Unfortunately, some aspects need to be corrected.

I will start from the beginning.

-          In the abstract, words are not separated by spaces at the beginning of the first sentence.

-          Latin names of species should be italicized - line 19, 278, 279 and all the references.

-          Line 25 and 26 the names of compounds should be written in lower case.

-          Line 64 in the cited work should be Ammar (capitalized, of course) instead of Sonda (name).

-          There are no standard deviations in the content of compounds in Table 2.

-          In Figure 1, all compound names should be written in the same way - uppercase or lowercase.

-          Table 3, row 1, column 3, it is better to write Molecular mass or weight rather than just mass; column 4 should be [M-H]-; compound number 3, should be written in English not French. Compound 14 [M-H]- instead of 632 it should be 623.

-          The correct name for compound 28 is nuezhenide or nuzhenide, also in line 188.

-          Figure 2 - in the description the letters A, B and C are meaningless.

-          Lines 124 and 141 - retention times should be given to the second decimal place.

-          Line 234 - MS-MS should be capitalized.

-          Line 300 - what compound was used for the calibration curve?

-          Line 346 - should be "did not" not "didn't".

-          The conclusions are too general.

Author Response

According to reviewer #3:

In the abstract, words are not separated by spaces at the beginning of the first sentence.

Answer: As the reviewer suggests, we have corrected.

Latin names of species should be italicized - line 19, 278, 279 and all the references.

Answer: As the reviewer suggests, we have corrected them.

Line 25 and 26 the names of compounds should be written in lower case.

Answer: As the reviewer suggests, we have corrected.

Line 64 in the cited work should be Ammar (capitalized, of course) instead of Sonda (name).

Answer: As proposed by the reviewer, we have changed “Sonda” to “Ammar”.

There are no standard deviations in the content of compounds in Table 2.

Answer: as suggested by the reviewer the standard deviations were added as all experiment has been repeated 3 times.

In Figure 1, all compound names should be written in the same way - uppercase or lowercase.

Answer: As the reviewer suggests, we have corrected.

Table 3, row 1, column 3, it is better to write Molecular mass or weight rather than just mass; column 4 should be [M-H]-; compound number 3, should be written in English not French. Compound 14 [M-H]- instead of 632 it should be 623.

Answer: As the reviewer suggests, we have corrected them.

The correct name for compound 28 is nuezhenide or nuzhenide, also in line 188.

Answer: As suggested by the reviewer, we have corrected them.

Figure 2 - in the description the letters A, B and C are meaningless.

Answer: Dear Reviewer, thank you for this remark, we have chosen to study each part of the root separately; we have divided it into three parts, the outer part, the inner part and the sum. Each part is represented by a chromatogram. As you suggested, we have improved this part of the BPC chromatogram discussion.

Lines 124 and 141 - retention times should be given to the second decimal place.

Answer: As suggested by the reviewer, we have corrected.

Line 234 - MS-MS should be capitalized.

Answer: As suggested by the reviewer, we have corrected.

Line 300 - what compound was used for the calibration curve?

Answer: Dear Reviewer, thank you for this question, the compound used for calibration is caffeic acid.

Line 346 - should be "did not" not "didn't".

Answer: As suggested by the reviewer, we have corrected.

The conclusions are too general.

Answer: Dear Reviewer, thank you for this remark, the conclusion section was revised and ameliorated.

Round 2

Reviewer 1 Report

The revised manuscript was sufficiently improved but there are some considerations:

Table 2: All the numbers used in the SD round them to two decimal places.

Table 3: Please correct TR to RT.

Please add a discussion for the first results part 2.1. (extraction yield)

The antiviral effects of hydroethanolic extracts of O. europaea were in contrast with literature, The authors did not give an explanation for these results. Differences might be due among others to differences in extraction methods used, in chemical characteristics of the plant used (depending on the growth conditions of the plants). This needs to be added as a possible explanation for the different results.

Please revise all figure captions in the manuscripts. The figure captions should contain a brief description of the experiments so that the figure can be understood without main text.

Line 336: please correct “foetal” to “fetal”

Line 342: please subscribe 2 in CO2

Line 347:  please change “Notably” or delete it.  

Please remember to write the abbreviation in full only at the first time in which it is reported for example DMEM

Please remember that abbreviations should only be used if the term appears two or more times. Revise along the manuscript. For example, DMSO

Author Response

view response to reviewer 1

Reviewer 2 Report

Please check the references. Some of the citations are not correctly typeset; e.g., talhaoui et al [13], and/or the year is missing; e.g., Ammar et al

Please homogenize the reporting of numbers. E.g., to use the same number of digits in tables. For example, 4.54 ±0.19; 2.227 ±0.1 are not using the same format.

Figure captions:  Comparison of the content of some compounds in roots and leaves (. missing): They are 3 compounds. You could write e.g. 'selected compounds' or 'relevant compounds'.

Please proofread the text again. E.g.: "In their chemistry, low molecular weight sugars present multiple properties in common. " => "Low molecular weight sugars share multiple chemical properties."

Please report mass-to-charge values correctly. E.g.  at (m/z 623.20) is either "at a mass-to-charge ratio of 623.20" or "623.20 m/z."

Several superfluous phrases can be eliminated, such as "In order to", and "indeed," and 'overselling' words such as 'tackle' could be avoided. 

Conventions on the reporting of numbers and units in scientific papers should be read and followed, e.g. the spelling-out of numbers at the beginning of sentences, the consistent reporting of digits, mass units, etc.

Author Response

view response to reviewer 2

Reviewer 3 Report

Changes have been made to this article but there is still a need for corrections.

In Table 2, it is enough if the results and standard deviation are given to the second decimal place.

Table 3, compound 39 and page 6, line 228, the name should be written "Olivil 4-O-β-D-glucopyranoside" instead of "Olivil 4-O-β-D-glucopyranoside".

There is still no detailed description of Figure 3. The description of Figure 3 lacks what the letters A, B and C stand for; yes, it is explained in the text, but it should also be in the description of the drawing.

Author Response

view response to reviewer 3
